# A Simplified Approach to Pulmonary Vein Visualization during Cryoballoon Ablation of Atrial Fibrillation

**DOI:** 10.3390/medicina58121700

**Published:** 2022-11-22

**Authors:** Omar Anwar, Da-Un Chung, Melanie A. Gunawardene, Christiane Jungen, Jens Hartmann, Christian Meyer, Nele Gessler, Stephan Willems, Samer Hakmi, Christian Eickholt

**Affiliations:** 1Department of Cardiology and Intensive Care Medicine, Asklepios Clinic St. Georg, Faculty of Medicine, Semmelweis University Campus Hamburg, 20099 Hamburg, Germany; 2DZHK (German Center for Cardiovascular Research), Partner Site Hamburg/Kiel/Luebeck, 20251 Hamburg, Germany; 3Department of Cardiac Electrophysiology, Heart and Vascular Centre, University Hospital Hamburg Eppendorf, 20251 Hamburg, Germany; 4Division of Cardiology, Cardiac Neuro- and Electrophysiology Research Consortium (cNEP), EVK Düsseldorf, 40217 Düsseldorf, Germany; 5Cardiac Neuro- and Electrophysiology Research Consortium (cNEP), Institute for Neural and Sensory Physiology, Medical Faculty, Heinrich Heine University Düsseldorf, 40225 Düsseldorf, Germany; 6Department of Cardiology, Klinikum Itzehoe, 25524 Itzehoe, Germany

**Keywords:** atrial fibrillation, cryoballoon, pulmonary vein isolation, angiography

## Abstract

*Background and Objectives*: Selective pulmonary vein (PV) angiography has been established as the gold standard for PV visualization in cryoballoon (CB)-based pulmonary vein isolation (PVI). We sought to simplify this approach to reduce procedural complexity and radiation exposure. *Materials and Methods*: Patients with paroxysmal and recently diagnosed persistent AF undergoing CB-based PVI from January 2015 to December 2017 were retrospectively analyzed. Patients underwent either selective PV angiography or conventional left atrial (LA) angiography for PV visualization. *Results*: A total of 336 patients were analyzed. A total of 87 patients (26%) received PV angiography and 249 (74%) LA angiography. LA angiography required fewer cine-sequences for PV visualization, translating into a significant reduction in procedure duration, fluoroscopy time and dose area product. Additionally, less contrast medium was utilized. PV occlusion by the CB, CB temperature and time to isolation showed no significant differences. The number of CB applications and total application time (LA angiography: 1.4 ± 0.02 vs. PV Angiography: 1.6 ± 0.05; *p* < 0.0001; LA angiography: 297.9 ± 4.62 vs. PV-Angiography: 348.9 ± 11.03; *p* < 0.001, respectively) per vein were slightly but significantly higher in the PV angiography group. We observed no difference in late AF recurrence (24.7% LA angiography vs. 21.3% PV angiography; *p* = 0.2657). *Conclusions*: A simplified protocol, using LA angiography for PV visualization, entails a reduction in procedure time and radiation exposure while equally maintaining procedural efficiency and safety in both groups.

## 1. Introduction

Pulmonary vein isolation (PVI) by cryoballoon (CB) ablation is an effective and efficient alternative to point-by-point radiofrequency ablation in patients with atrial fibrillation (AF) [1]. It requires less technical infrastructure (e.g., absence of a 3D mapping systems) [2] and typically has a steeper learning curve. Selective, multiplanar pulmonary vein (PV) angiography has been established as the gold standard for PV visualization in CB PVI [3,4,5]. We sought to simplify this approach to reduce procedural complexity and radiation exposure.

## 2. Methods

Patients with paroxysmal and recently diagnosed persistent AF undergoing CB-based PVI at our institution from January 2015 to December 2017 were retrospectively analyzed. The aim of this study was to compare two different approaches (LA angiography vs. selective PV angiography) for fluoroscopic PV visualization in CB-based PVI, assessing the impact on radiation exposure, procedural safety and acute, as well as long-term efficacy. This study was performed in compliance with the Declaration of Helsinki and was approved through our local ethics Committee (Ärztekammer Hamburg, 2001-300084-WF).

### 2.1. Procedural Setting

Visualization of the left atrium/pulmonary veins prior to the procedure via computed tomography or magnetic resonance imaging was not performed. 

All patients provided written informed consent. Intracardiac thrombus formation was excluded by transesophageal echocardiography prior to the procedure. For the duration of the procedure, deep sedation was maintained by continuous infusion of propofol and bolus administration of fentanyl, under continuous monitoring of vital parameters (ECG, SpO2, NIBP).

After femoral venous access, a steerable octapolar diagnostic catheter (Inquiry 6F, St. Jude Medical, St. Paul, MN, USA) was placed in the coronary sinus (CS). Intracardiac electrograms were acquired via an electrophysiology recording system (LabSystem Pro, Bard Electrophysiology, Lowell, MA, USA). For cardiac pacing and phrenic nerve stimulation, a programmable stimulation device (UHS 3000 or QubicStim, Biotronik, Berlin, Germany) was used. LA access was achieved via transseptal puncture using a standard sheath and needle combination (SL-0, 8F and BRK-1 needle, St. Jude Medical, St. Paul, MN, USA) under fluoroscopic control. Periprocedural anticoagulation with unfractionated heparin was initiated after transseptal puncture (100 IU/kg bolus) and titrated to an activated clotting time (ACT) > 300 s. LA sheaths were irrigated with heparinized saline solution at a flow rate of 2 mL/h.

### 2.2. Pulmonary Vein Visualization and Cryoballoon Guidance

Patients received either conventional selective PV angiography with contrast injection into the PV via a 7F NIH angiography catheter (Cordis, Miami Lakes, FL, USA) or an unselective LA angiography for PV visualization based on operator preference. LA angiography was performed by manual contrast injection (20 mL per injection) in the AP projection via the SL-0 sheath (St. Jude Medical, Saint Paul, MN, USA) used for transseptal access [6]. Both PV pairs were visualized with one injection by rotating the SL-0 sheath counter-clockwise (Figure 1). Additional injections were performed in case the LA/PV contrasted poorly. For the LA angiography group, the anterior-posterior (AP) plane was maintained during the remaining CB procedure (Figure 2). The fluoroscopy plane was adjusted only in case of inadequate PV occlusion. For PV angiography, the NIH catheter was placed into the LA via the aforementioned SL-0 sheath (St. Jude Medical, Saint Paul, MN, USA).

Each PV was then selectively intubated and angiographically displayed (5 mL per injection, IMERON 350 mg/mL, Bracco Imaging, Milano, Italy) in an oblique projection (left PVs = LAO 30°, right PVs = RAO 30°), to achieve optimal visualization of PV anatomy. Additional injections in complmentary projections were added when necessary. Intracardiac echocardiography and/or a 3D-mapping system was not utilized in either approach.

### 2.3. Cryoballoon Ablation

After fluoroscopic visualization of the PVs, the SL-0 sheath was replaced with a 12F steerable sheath (FlexCath Advance, Medtronic, Minneapolis, MN, USA) and flushed continuously with heparinized saline solution (2 mL/h) to prevent thrombus formation. Subsequently, a 28-mm CB (ArcticFront Advance, Medtronic, Minneapolis, MN, USA) was positioned in the LA, through which an octapolar spiral mapping catheter (Achieve 15 mm or 20 mm, Medtronic, Minneapolis, MN, USA) was advanced. The PVs were then sequentially intubated using the mapping catheter and occluded with the inflated CB. To assess the quality of the occlusion, contrast injection over the distal tip of the CB was performed (3–5 mL per injection) via a dedicated injection pump (ACIST CVi, Bracco Imaging, Milan, Italy) and graded on a scale from 1 (insufficient occlusion with substantial escape of contrast agent) to 4 (perfect occlusion with no leakage of contrast agent) [7]. Recording of PV signals by optimal placement of the mapping catheter during PV occlusion was attempted, if necessary by repositioning of the spiral mapping catheter. During ablation, esophageal temperature was monitored with a multipolar esophageal temperature probe (CIRCA-S-CATH, Circa Scientific, Englewood, CO, USA or SensiTherm, St. Jude Medical, St. Paul, MN, USA). To monitor phrenic nerve function during ablation of the right sided PVs, pacing was performed via the octapolar mapping catheter, relocated from the CS to the superior vena cava for this purpose, with recording of the diaphragmatic compound motor action potential (CMAP) via the surface ECG [8]. The general target ablation time was 240 s, which was adjusted in case of an inefficient freeze, a rapid decline in temperature (<−30 °C within 30 s), a steep decline in temperature (>−55 °C) or any sign of collateral tissue damage (phrenic nerve impairment, esophageal temperature >−17 °C). Additional freezes were delivered in cases of failed PV isolation or, at the performing physician’s discretion, in cases of delayed electric pulmonary vein isolation (>120 s) or inadequate CB temperature (<−30 °C). PV isolation was confirmed immediately after ablation as well as at the end of the procedure, via the spiral mapping catheter. 

### 2.4. Statistics

Continuous variables are presented as mean ± standard error of the mean; absolute and relative frequencies are given for categorical data. Continuous variables were compared using Student’s *t*-test and categorical variables were compared using Fisher’s exact test. A *p*-value < 0.05 was considered statistically significant. Statistical analyses were performed using GraphPad Prism 6.0 (Version 9.4.1) for Macintosh (GraphPad Inc., La Jolla, CA, USA).

## 3. Results

Patient characteristics: A total of 336 patients were analyzed (62 ± 0.6 years; 127 women [38%]; BMI 27 ± 0.2 kg/m^2^; LA volume 66 ± 2.4 mL; EHRA score 2.3 ± 0.04; CHA_2_DS_2_-VASc score 2.0 ± 0.1). A total of 87 patients (26%), enrolled between January 2015 and December 2017, received PV angiography and 249 patients (74%) underwent LA angiography, with both groups showing no significant differences in baseline characteristics (Table 1).

Radiation exposure: LA angiography required fewer cine-sequences for PV visualization, translating into a significant reduction in procedure time, as well as a lower fluoroscopy time and consequently a reduction in radiation exposure. Additionally, there was a trend towards a lower volume of contrast agent used with LA angiography (78.7 ± 2.8 vs. 91.3 ± 5.99 mL; *p* = 0.057) (Figure 3).

Procedural parameters: Procedural variables, such as the quality of PV occlusion by the CB, CB temperature and time to isolation showed no significant differences (Table 2). Out of the 249 patients in the LA angiography group, 55 (22%) patients received a single injection, 154 (61%) patients required two injections for LA/PV visualization and 40 (16%) patients required more than two contrast medium applications. The number of CB applications (1.4 ± 0.02 vs. 1.6 ± 0.05; *p* < 0.0001) and total application time per vein (298 ± 4.62 vs. 349 ± 11.03 s; *p* < 0.0001) were slightly but significantly higher in the PV angiography group.

Safety: Complications consisted mainly of minor bleeding events; they were equally distributed between both groups. One major complication (stroke) was observed in the PV angiography group (0.01%). A total of four patients (LA angiography: n = 2, 0.008% and PV angiography: n = 2, 0.02%) suffered a phrenic nerve injury, one of which resulted in permanent phrenic nerve damage. The remaining three phrenic nerve injuries resolved within 6 months.

Long-term follow-up: After a mean follow-up period of 354 days (initial blanking period of 90 days), no significant difference with regard to late recurrence of AF could be observed (PV angiography n = 17, 21.3% vs. LA angiography n = 53 patients, 24.7%; *p* = 0.2657) (Figure 4).

## 4. Discussion

While the technology and application of CB-PVI have undergone significant development over the last years, including advances in balloon design and elimination of the routine use of a bonus freeze, as well as dynamical dosing schemes adapting to the effect of each freeze cycle, no relevant changes have been implemented concerning the periinterventional visualization of the pulmonary vein anatomy. While different pre- (computed tomography [9,10,11], magnetic resonance imaging [12]) and peri-interventional methods (e.g., non-fluoroscopic [13,14,15], rotational angiography [16,17] transesophageal- or intracardiac ultra-sound [10,18]) [4] have been employed or proposed before or during PVI, their use has not been translated into routine CB PVI procedures in the majority of centers that do not utilize the above mentioned techniques. Selective PVA thus has remained the staple of periinterventional PV visualization and is recommended as the de facto gold standard [3,5,19].

### 4.1. Main Findings

The main findings of this study are as follows: (1) LA angiography required fewer cine-sequences for PV visualization, resulting in a reduction in total procedure and fluoroscopy time, as well as a reduction in dose area product; (2) No significant differences were observed in predictors of procedural success, such as quality of PV occlusion, CB temperature and time to isolation in both groups; (3) No significant differences were observed in recurrences of atrial fibrillation in both groups during long-term follow-up.

#### 4.1.1. LA Angiography Is Safe

Neither peri-interventional nor post-interventional complications were observed with the LA angiography approach. Practically, the use of a single SL-0 sheath eliminates the need for additional positioning of an additional angiography catheter and minimizes manipulation in the PV, which might be associated with an increased risk of mechanical complications. Furthermore, eliminating the exchange of an additional catheter may reduce further risk of an air/thromboembolism. With a standardized, single fluoroscopic projection, the gross anatomy of the pulmonary veins, as well as the LA, can be appreciated and leveraged for optimal CB placement. This also pertains to the identification of an accessory- or anatomically deviant PV, which may escape the operator’s attention using selective angiography [19]. Although the PV ostia have been the initial focus of pulmonary vein isolation for many years, the shift to more antral lesions of the pulmonary veins does not only reduce the risk of pulmonary vein stenosis but also reduces the need for precise anatomy and localization of the PV ostia [11,19,20]. Imnadze et al. [21] previously proposed a similar approach. In this step-wise visualization of the PV via LA angiography, contrast medium was injected via an SL-1 Sheath (SL-1, St. Jude Medical, St. Paul, MN, USA) during rapid ventricular pacing (via an additional diagnostic catheter in the right ventricular apex) in order to minimize atrial emptying. In our study, sufficient information of LA/PV anatomy was acquired without the use of additional maneuvers or diagnostic catheters. A close relationship between the amount of contrast medium, speed of injection and LA size exists, however, the relationship of these parameters were not analyzed in this study.

#### 4.1.2. LA Angiography Leads to a Reduction in Radiation Exposure and Is More Efficient

Fluoroscopy times in this study in both groups were in line with current literature [1]. Due to the standardized, single fluoroscopic projection in the LA angiography approach, a reduction in fluoroscopy time and a reduction in dose area products were observed, thus reducing the radiation exposure of the patient and the operator alike. The omission of the multi-purpose catheter during the PVA approach and elimination of multiple fluoroscopic projections also leads to a more efficient cryoballoon procedure in terms of total procedure time (Table 2).

#### 4.1.3. LA Angiography Is Effective

In all CB procedures, the exact position and depth of the inflated cryoballoon is assessed in relation to the PV ostium with contrast medium before freezing, making a prior assessment of specific PV anatomy seem redundant to some extent. Interestingly, in this study, no difference in quality of CB occlusion of the PV, temperature of the CB and time to isolation was observed in either group. The effectiveness of the LA angiography approach is reflected during the follow-up of 294 patients (see “Follow-up”), which shows no significant difference in terms of recurrence of atrial fibrillation, rehospitalization or death.

#### 4.1.4. Clinical Implications

A standardized, single fluoroscopic projection during angiography of the LA seems sufficient to visualize the LA/PV and can lead to a reduction in fluoroscopy time, radiation exposure and a reduction in procedure time. This contributes to the resolution of one of the main limitations of CB PVI; the exposure of the patient to still relevant amounts of contrast agent and radiation. This might facilitate the use of this effective and efficient mode of AF treatment in a wider selection of patients.

### 4.2. Limitations

Firstly, this study was a nonrandomized study and not designed to compare procedural outcomes in terms of AF recurrence. In addition, the sample size in the PV angiography group is smaller in comparison to the LA angiography group. Our findings, therefore, may require verification in a lager, randomized prospective study. Furthermore, this study focuses on centers that do not utilize pre- or periinterventional imaging techniques for LA/PV visualization, as specialized centers around the world are employing non-fluoroscopic approaches with complete omission of fluoroscopic visualization of the LA or PV [22]. Furthermore, a close relationship between the amount of contrast medium, speed of injection and LA-Volume exists, however, this study was not designed to assess the relationship of these parameters.

## 5. Conclusions

A simplified CB PVI protocol, using LA angiography for PV visualization, entails a reduction in procedure time and radiation exposure while equally maintaining procedural quality, efficiency and safety in both groups.

## Figures and Tables

**Figure 1 medicina-58-01700-f001:**
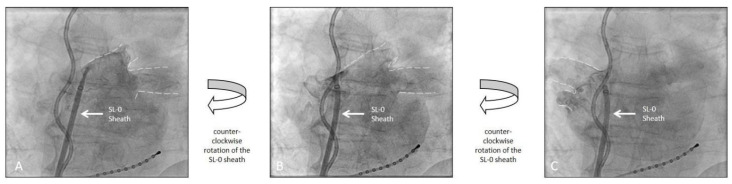
Left atrial angiography demonstrating the visualization of the pulmonary veins and contour of the left atrium with a single injection of contrast medium. (**A**): initial injection of contrast dye for the left pumonary veins (dotted lines). (**B**): Continuation of the initial injection of contrast dye with a counter-clockwise rotation of the SL-0 Sheath further appreciating the anatomy of the left pulmonary veins (dotted lines). (**C**): Continuation of the initial injection of contrast dye with an additional counter-clockwise rotation now visualizing the right pulmonary veins (dotted lines). The fluoroscopic images were recorded in an anterior-posterior projection (AP). (Also see Appendix A).

**Figure 2 medicina-58-01700-f002:**
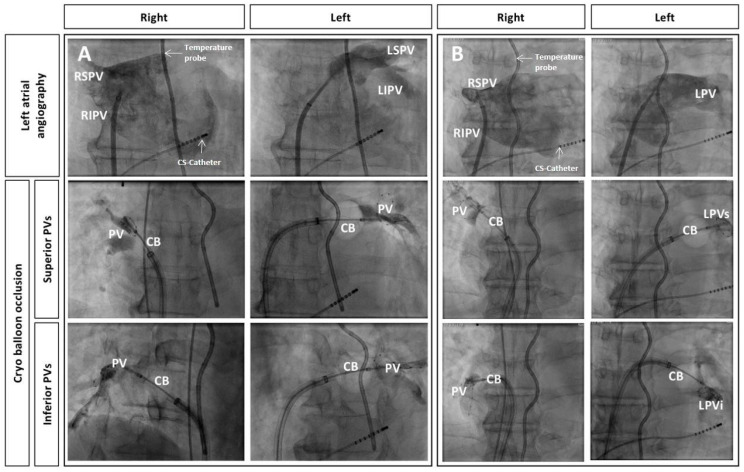
Left atrial angiography and subsequent occlusion of pulmonary veins (PVs) by cryoballoon (CB) placement, all in anterior-posterior projection. Additionally depicted are the multipolar esophageal temperature probe and the octapolar diagnostic catheter placed in the coronary sinus. (**A**): Normal anatomy, (**B**): Left-sided common ostium (LPV) with superior-inferior (LPVs, LPVi) branches; (LSPV: left superior PV, LIPV: left inferior PV, RSPV: right superior PV, RIPV: right inferior PV).

**Figure 3 medicina-58-01700-f003:**
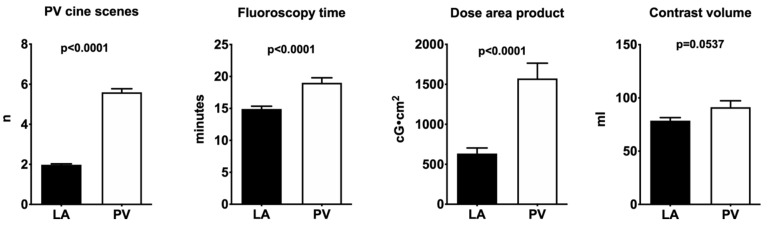
Graphical depiction of the results (PV: pulmonary veins; LA: left atrial angiography).

**Figure 4 medicina-58-01700-f004:**
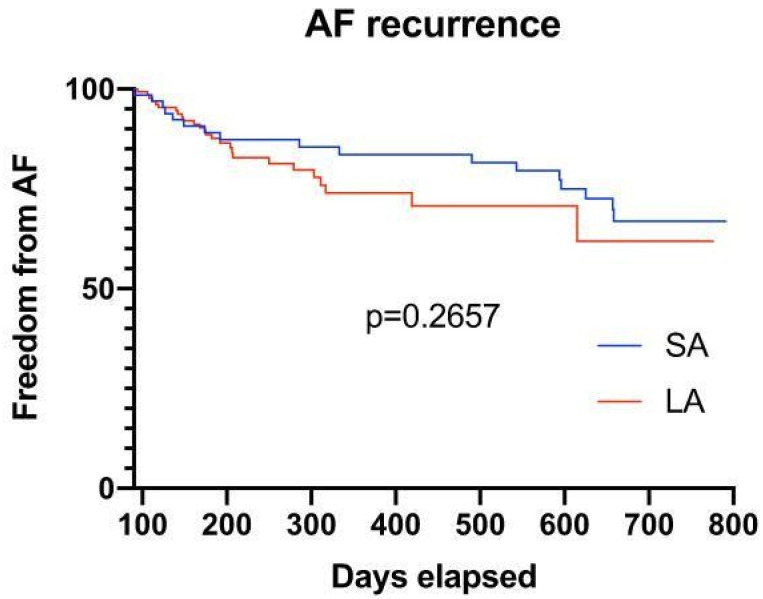
Survival plot of patients after a mean follow-up of 354 days, showing no significant difference with regard to late recurrence of AF in both groups (after a blanking period of 90 days).

**Table 1 medicina-58-01700-t001:** Baseline patient characteristics.

	LA Angiography (n = 249)	PV Angiography (n = 87)	
Age (y)	62.7 ± 0.69	60.9 ± 1.04	*p* = 0.16
Sex (female), n (%)	96 (39)	31 (36)	*p* = 0.70
BMI	27.2 ± 0.25	27.1 ± 0.40	*p* = 0.79
CHA_2_DS_2_-Vasc score	2.0 ± 0.09	1.8 ± 0.15	*p* = 0.15
Left atrial volume (mL)	70.8 ± 2.7	69.1 ± 4.8	*p* = 0.77
Paroxysmal atrial fibrillation, n (%)	68 (27)	19 (21)	*p* = 0.39
Coronary artery disease, n (%)	41 (17)	8 (9)	*p* = 0.11
Congestive heart failure, n (%)	10 (4)	4 (5)	*p* = 0.76
Valvular disease, n (%)	21 (8)	3 (4)	*p* = 0.15
GFR < 30 mL/min, n (%)	0 (0)	1 (1)	*p* = 0.22
Diabetes mellitus, n (%)	17 (6.9)	9 (10)	*p* = 0.35

BMI: body mass index; GFR: glomerular filtration rate.

**Table 2 medicina-58-01700-t002:** Procedural parameters.

	LA Angiography (n = 249)	PV Angiography (n = 87)	
Procedure duration (min)	93 ± 1.79	110.8 ± 3.18	*p* < 0.0001
Cryo applications *	1.4 ± 0.02	1.6 ± 0.05	*p* < 0.0001
PV occlusion quality scoring *; **	3.7 ± 0.02	3.8 ± 0.03	*p* = 0.14
Realtime recordings *	348 (35,8%)	129 (37,5%)	*p* = 0.60
Cumulative application time (min) *	297.9 ± 4.62	348.9 ± 11.03	*p* < 0.0001
Time to isolation (min) *	62.8 ± 2.56	53.4 ± 3.92	*p* = 0.053
Temperature at isolation (°C) *	−34.1 ± 0.53	−33.1 ± 1.01	*p* = 0.35
Minimum balloon temperature (°C) *	−46.2 ± 0.25	−46.8 ± 0.43	*p* = 0.22

* per single pulmonary vein; total number of veins: 972 (LA angiography), 344 (PV angiography). ** from 1 (insufficient occlusion with large efflux of contrast medium) to 4 (best occlusion with no efflux of contrast medium), see text for details.

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
