# Peer review of "A Simplified Approach to Pulmonary Vein Visualization during Cryoballoon Ablation of Atrial Fibrillation"

_medicina, 2022, doi:10.3390/medicina58121700_

Round 1

Reviewer 1 Report

This is an interesting retrospective study regarding visualization of pulmonary veins (PV) by simple LA angiography in the cathlab during PV cryoablation. In this proposed (new) technique the authors performed direct manual contrast injection in the LA via the sheath used for trans-septal puncture in order to properly identify the ostia and the final segments of PVs. The current alternative technique uses separate incannulation of every PV with a dedicated NIH catheter; this is associated with prolonged procedural times and increased radiation exposure. 

All patients included in the retrospective analysis did not have preprocedural angio-CT scans to identify the PVs, nor intracardiac ultrasound was used during the diagnostic phase of the ablation procedures. This should also be an economical gain for patients submitted to an already expensive and lengthy procedure.

As expected performing unselective PV visualization by LA angiography resulted in lower procedural time and radiation exposure compared with selective PV angiography. As a consequence the authors propose the use of LA angiography as a useful technique for initial PV identification prior cryoballoon ablation in paroxysmal AFib.

The aim of the study and working hypothesis are sound. Description of methods is thorough. The results are clearly stated and commented in the discussion section. Study groups are described properly both for general characteristics and procedural data. Angiographic images are clear enough and the quality is acceptable. References are updated and sustain text.

As a minor comment, I would sugest to authors to comment on the relationship between LA volume assessed by transthoracic echocardiography and the volume of contrast manually injected in the LA. Apparently all patients received 20 ml of contrast injected by hand, irrespective of LA volume. There is a close relationship between the volume of the chamber/vessel an operator wants to opacify and the volume/speed of contrast administration to obtain good angiographic data. I would wellcome a short comment on that.   

Otherwise, I agree to publication of this paper as is.      

Author Response

Comment:

"...There is a close relationship between the volume of the chamber/vessel an operator wants to opacify and the volume/speed of contrast administration to obtain good angiographic data. I would wellcome a short comment on that." 

Dear Reviewer 1,

we first and foremest thank you for your positive comments and immaculate review of our article. We made following changes to our manuscript:

On page 9 under the heading of "LA angiography is safe" we added the following comment: "A close relationship between the amount of contrast medium, speed of injection and LA size exists, however, the relationship of these parameters were not analyzed in this study."

Additionally, we added the following comment to the "Limitations"-section (page 10) of this manuscript: "...Furthermore, a close relationship between the amount of contrast medium, speed of injection and LA-Volume exists, however, this study was not designed to assess the relationship of these parameters."

We again thank you, Reviewer 1, for your time and effort.

Kind regards,

Omar Anwar

Reviewer 2 Report

A simplified approach to pulmonary vein visualization during cryoballoon ablation of atrial fibrillation

Manuscript entitled “A simplified approach to pulmonary vein visualization during cryoballoon ablation of atrial fibrillation” by Anwar et al., is a good study. Here, the authors developed a simplified protocol, using left atrial angiography for the selective pulmonary vein visualization entails a reduction in procedure time and radiation exposure while equally maintaining procedural efficiency and safety. The left atrial  angiography required fewer sequences for pulmonary vein visualization, resulting in a reduction in total procedure and fluoroscopy, as well as a reduction in dose area product.

Overall, the information presented in this research article is a useful approach to visualize the pulmonary vein. The methods and results presented in the manuscript are clear/easy to understand. I approve its publication. 

Author Response

Dear Reviewer 2,

thank you for your immaculate review, kind words and positive comments!

Kind regards,

Omar Anwar